# Effects of Habitat Loss on the Ecology of *Pachyphytum caesium* (Crassulaceae), a Specialized Cliff-Dwelling Endemic Species in Central Mexico

Ricardo Clark-Tapia [1], Gabriel González-Adame [1], Jorge E. Campos [2], Victor Aguirre-Hidalgo [1], Nelly Pacheco-Cruz [2], Juan José Von Thaden Ugalde [1], Samuel Campista-León [3], Luz Isela Peinado-Guevara [3] and Cecilia Alfonso-Corrado [2,*]

[1] Laboratorio de Estudios Ambientales, Universidad de la Sierra Juárez, Avenida Universidad s/n, Ixtlán de Juárez, Oaxaca 68725, Mexico; rclark@unsij.edu.mx (R.C.-T.); gaboadame@unsij.edu.mx (G.G.-A.); victor@unsij.edu.mx (V.A.-H.); juanvonthaden@gmail.com (J.J.V.T.U.)

[2] Unidad de Biotegnología y Prototipos, Facultad de Estudios Superiores de Iztacala, Universidad Nacional Autónoma de México, Avenida de los Barrios 1, Los Reyes Iztacala, Tlaneplantla 54090, Mexico; jecampos@me.com (J.E.C.); n311pc@gmail.com (N.P.-C.)

[3] Laboratory of Microbiology and Applied Biology, Faculty of Biology, Universidad Autónoma de Sinaloa, Avenida Universitarios s/n, Ciudad Universitaria, Culiacán Rosales 80013, Mexico; samcl@uas.edu.mx (S.C.-L.); luzipg@uas.edu.mx (L.I.P.-G.)

\* Correspondence: liana@unsij.edu.mx

**Abstract:** Cliff-dwelling plant species are highly specialized and adapted to a vulnerable, fragmented, and are mostly endemic, narrowly-distributed and threatened. As a contribution to the conservation efforts of endemic cliff-dwelling species, this study provides an overview of the effects of habitat loss on the abundance and distribution of *Pachyphytum caesium* (Crassulaceae) due to human disturbances. To achieve this objective, we first conducted a retrospective analysis from 2003–2013 to assess the effects of land use change on the abundance of *P. caesium*. Secondly, we estimate the abundance and distribution of *P. caesium* throughout the study area, as well as analyze the effect of rock-climbing activities on the density and population structure of *P. caesium*. The results suggest differences in population abundance among sites is due to the adverse effects of habitat loss. *P. caesium* presents a very restricted distribution with small and fragmented populations. In addition, guava agriculture has a significant impact on the chemical soil properties of the hillsides, causing a significant effect on the occurrence of *P. caesium*, while sport activities remove both the soil and the plants from the cliffs. According to the results, *P. caesium* is classified as a plant species with extremely small populations (PSESP), and it is highly vulnerable to habitat disturbance. Its conservation is thus a priority to ensure its permanence.

**Keywords:** abundance; climbing; fragmentation; land-use change; *Pachyphytum*; soil properties; size structure

## 1. Introduction

In recent decades, the abundance and distribution of species have been affected by climate change and habitat loss, which have increased the vulnerability and the possibility of extinction in several plant species [1–4]. Currently, this problem is the critical in plant species with extremely small populations (PSESP), a concept originated in China to guide the conservation of rare and endangered plant species [5,6]. PSESP are rare, endemic, and endangered species with small population sizes, specialized habitat, restricted distribution, and low genetic variation. Consequently, PSESP are highly vulnerable to human disturbances and environmental changes, which make them susceptible to extinction [6–10]. In the face of the global biodiversity crisis due to the mass extinction of many species [11], our knowledge about the factors that affect PSESP are a priority for developing suitable conservation strategies [10].

Survival of PSESP inhabiting human disturbed areas is more sensitive to the presence of invasive exotic species, overexploitation, and illegal trade [7,9,11], as well as fragmentation and loss of habitat [4,7,12]. In particular, fragmentation and habitat loss are considered global change forces with negative effects in microhabitats [12]. Land use change generates highly modified landscapes in which the predominant matrix is usually composed of agricultural fields and urban settlements and, to a lesser extent, an area of preserved vegetation [12]. However, the magnitude of this impact can vary greatly depending on the removal method of vegetation cover, the type of land use, soil management, and cropping/livestock systems, among other factors [13].

Cliff-dweller plants are highly specialized and vulnerable species to extinction. This is because most cliffs (an area of vertical rock exposure, or nearly vertical, made up of a platform, a rock base pediment, and a cliff-face with rock exposure in between) are extremely unstable and fragmented habitats that make life difficult [14]. Therefore, any environmental change (e.g., rock chemistry, loss of nutrients, and soil removal due to water erosion) in the cliff can affect the structure and function of cliff-dwelling species [14,15]. In addition, the effect of disturbances on cliff platforms due to agriculture (e.g., changes in vegetation cover and soil chemistry) can increase vulnerability to species extinction; however, these effects have not been explored much in species located on cliffs. Another human threat that affects cliff species is rock-climbing activities, which negatively impact the face and affect the edge and base, causing loss of soil and direct damage to the entire plant or to some part of it [14–17]. Although rock-climbing on cliffs has a recent history in central Mexico, its popularity has increased dramatically since the 1990s, and at present, it is a usual ecotourist activity, of which its ecological impact is also unknown.

*Pachyphytum caesium* Kimnach and Moran (Crassulaceae) is a rare and micro endemic species discovered in 1993 [18]. Currently, its biology, ecology, and genetics are unknown; however, this aspect is not exclusive to *P. caesium*, as, of the 25 species of the genus *Pachyphytum*, only their taxonomy is completely known, and they are also succulent hanging plants that inhabit cliffs or canyons [19,20]. Particularly regarding *P. caesium*, there is only a report of two isolated populations with a small population size that are restricted to cliffs in the remnants of dry tropical forests, and it transitions to oak forests, in the state of Aguascalientes [18,19]. This habitat loss is a problem for *P. caesium* because approximately 50% of these forests have been modified by human activities [21], making this natural habitat one of the most threatened worldwide [21,22]. In particular, certain regions of central Mexico have been transformed in more than 95% of its original area to agriculture and livestock land use [23]. However, the level of change where the species is located is unknown.

In order to contribute to the conservation efforts of cliff-dwellers, this study provides an overview of the effects of habitat loss in the distribution and environmental vulnerability of *P. caesium*. Therefore, the objectives of this study were as follows: (1) to evaluate the effect of land use changes between 2003–2013 on the abundance of *P. caesium*, (2) to assess the abundance, density, and size structure in all sampling sites, (3) to determine the effect of rock-climbing activities on the abundance and size structure of the population of the species, (4) to analyze the physical and chemical characteristics of the soils in sites with a presence and absence of the species, and (5) to determine if *P. caesium* can be classified as a PSESP and to provide strategies for its conservation.

## 2. Materials and Methods

### 2.1. Study Area

Our study area encompassed a region located between 18°51′ to 25°6′ N latitude and 103°25′ to 102°11′ W longitude, a central region in Mexico (in the states of Aguascalientes, Jalisco, and Zacatecas). It has a territorial area of 152,491 km$^2$ (Figure 1). Three major types of climates are distinguished in the study area: arid semi-warm (40%), semi-arid warm (25%), and temperate sub-humid (18%). Two climates: Warm sub-humid, and semi-cold sub-humid climates are present in less than 10% in the study area [24]. The

Annual temperature ranged from 8 to 29.5 °C. Elevation varies considerably from 1400 to 3050 masl [25]. The annual average precipitation is 875.6 mm and the evaporation is 780 mm year$^{-1}$ [24].

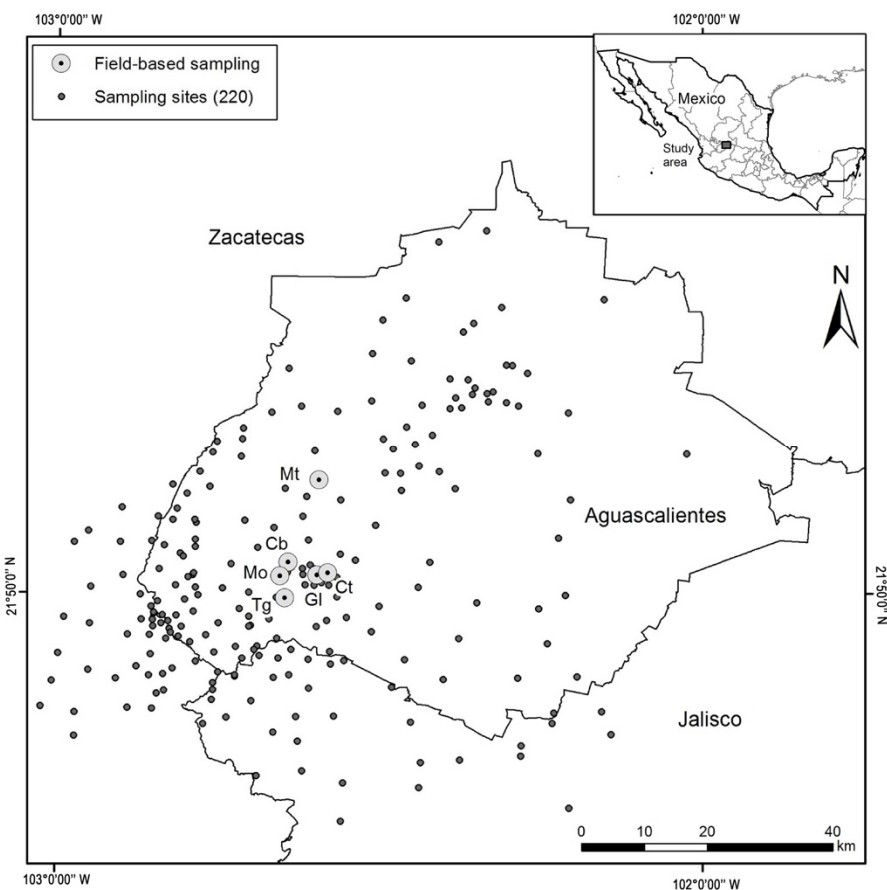

**Figure 1.** Study area in the central region of Mexico (220 sampling sites) and distribution of field-based sampling in the Aguascalientes State. Mt: Barranca Montoro; Cb: Presa Cebolletas; Mo: Presa Malpaso; Tg: Barranca Tortugas; Gl: Río Gil; and Ct: Puente Cuates.

*2.2. Species Description*

   *Pachyphytum caesium* (Crassulaceae) is a succulent perennial plant growing in the north-central region of Mexico [19]. The species has a restricted distribution to rocky cliffs in tropical dry forest and its transition with oak forest [18,19]. The stems are commonly branched basally and are erect in young plants but elongated and pendulous with age, reaching 30 to 40 cm in length and 1.5 to 2 cm in thickness. Leaves form lax terminal rosettes; are grayish to grayish purple and, obovate-oblong, have measurements of 3 to 7 cm long, 2 to 3.5 cm wide, and 0.8 to 1.2 cm thick; and are slightly convex. Inflorescence of axilar cincinnus is usually unbranched and rarely divided, ca. 30 cm length, with 6 to 20 flowers, [18,26]. The petals are greenish cream with deep pink ventrally. This species presents two floral periods per year.

   The reproductive biology of *Pachyphytum caesium* is unknown; for this reason, reproductive information comes from *Echeveria* sp., a neotropical genus with morphological similarities that is phylogenetically close to *Pachyphytum* [27]. Pollen grains are prolate-spheroidal and tricolporated, and their viability is variable between species. Stigmatic receptivity varies with the ambient temperature, but is greater between 12:00 p.m. and 14:00 p.m. Thus, they have a self-compatible reproductive system; the allogamous system or cross-pollination provided by insects or birds shows a high pollinator limitation. Furthermore, minimal pollen movement is reported. In general, the plants produce a large number of infertile or vain seeds [28,29].

### 2.3. Land-Use and Land-Cover Classification (Lulc), Sampling Procedures and Deforestation Rate

Land use/land cover (LULC), maps were created from Landsat imagery (30 m spatial resolution) obtained from the USGS (https://earthexplorer.usgs.gov/, accessed on 20 August 2020) for the years 2003 and 2013. Both images were obtained with geometric corrections and an atmospheric correction (Level 2, Landsat). The images were classified with a Maximum Likelihood algorithm using 6 of the 7 bands (numbers 1–5 and 7). In this study, three LULC categories were identified: forest (woody vegetation), No Forest (pasture/crops), and water. An average of 50 sampling units was used for each LULC category (150 in total) derived from field visits (70) and higher resolution images available from Google Earth (80) in more inaccessible areas. To validate the final maps, 150 additional reference sampling units, from those used in the classification of images (50 from each category; 15 obtained in the field and 35 in Google Earth for hard-to-reach areas) were used in order to generate an area-based error matrix, as well as a kappa index, for each classification [30].

A buffer analysis was used to document the LULC change in 2003 and 2013 [31]. We generated buffers with extents of 500 m, 1 km, and 1.5 km to determine the optimal size buffer. Each one quantified the compositional (Shannon and Simpson diversity index) and configurational heterogeneity (mean size of the forest) using Fragstats [32]. The magnitude and tendencies of forest cover change were assessed by cartographic overlaps and by estimating the differences in forest cover between dates; for this, deforestation rates (*r*) were estimated using the equation proposed by the FAO [33]:

$$r = 1 - \left(1 - \frac{A_1 - A_2}{A_1}\right)^{1/t} \tag{1}$$

where $A_1$ is the forest area at $t_1$ (initial area), $A_2$ is the forest area at $t_2$ (final area), and $t$ is the time difference between $t_2$ and $t_1$ in years.

### 2.4. Fragmentation Metrics

To determine if the sampling sites presented fragmentation in either 2003 or 2013, we used Fragstats [32]. However, as many landscape metrics are redundant and statistically correlated, we chose a selected parsimonious set of uncorrelated metrics using a Spearman's Rank Correlation Coefficient (Rs) threshold of Rs > 0.6 (indicating a moderate correlation) [34], which described landscape configuration (e.g., area and shape) and aggregation [35]. The three metrics chosen for this study were patch density (PD), edge density (ED), and core percentage of the landscape (CPLAND). PD is the number of patches, ED is the percentage of the total landscape area occupied by the class edges, and CPLAND is the percentage of the total landscape area occupied by the core areas of the class using 100 m edges to define areas of core habitat in the target class [36]. According to this, forest landscapes with a greater density of patches, more edge, and less core area were assumed to indicate increased fragmentation [37].

### 2.5. Ecological Distribution, Abundance, and Size Structure

The potential distribution of *P. caesium* was used as a baseline to establish 220 sampling sites between 2005 and 2012. The sites were selected because they presented potential habitats (cliffs, steep slopes, and canyons) for the occurrence of the species. More sampling effort (150 sites) was made in Aguascalientes, because the type locality is located in the state [18], while in Jalisco and Zacatecas, 30 and 40 sites were sampled, respectively.

The abundance and size structure were estimated in six sites (Barranca Tortugas, Barranca Montoro, Río Gil, Presa Malpaso, Presa Cebolletas, and Puente Cuates) situated in Aguascalientes. These sites showed different altitudes, orientations, and dominant plant associations (Table S1). In sites where *P. caesium* was recorded, we established six systematic 25 m² (5 × 5 m) plots separated by 10 to 20 m. Additionally, to evaluate the effect of rappelling on the size and number of individuals, three more plots were established

directly on the practice route in sites with rock-climbing activities. However, the intensity and frequency of rappelling were not considered in this study. In each plot, every individual was measured (supported by climbing techniques) and then classified in five size categories according to total stem length: (1) seedlings 0.1–2.0 cm, (2) juveniles 2.1–4 cm, (3) non-reproductive adults 4.1–8 cm, (4) reproductive adults (1), 8.1–12 cm, (5) reproductive adults (2) >12 cm. To categorize individuals, we considered the length of each individual based on annual growth and their reproductive stage (>8 cm are reproductive individuals) obtained through a demographic study (Clark-Tapia unpublished results). Differences in the size and structure among sites were compared with a Kruskal-Wallis one-way by ranks. To understand the relationship between the abundance and the deforestation rate, they were correlated with the Spearman correlation coefficient [34].

The heatmap technique was used to visualize similarities between the sites and the relative abundance of the size categories. The heatmap is a two-dimensional technique for high-dimensional data initially used in genetic studies of microbial diversity (e.g., [38]). The resulting color-coded mosaic map and adjacent dendrogram indicate a functional relationship among size categories and sites. The numerical values are shown by colors, grouped in rows and columns by their similarity, represented by a cluster dendrogram of UPGMA with a Cophenetic correlation of 0.91. All of the statistical analyses were performed in R v. 3.5.3 [39].

### 2.6. Environmental Characterization of Sites and Soil Analysis

Seven environmental characteristics were recorded at each site, namely: orientation obtained by a compass (SUUNTO), electrical conductivity (EM), and pH (potential of hydrogen), which was assayed in a soil-paste extract saturation [40] using an OAKTON model PC700), and the percentage of organic matter (OM) according to Black [41]. Additionally, altitude, latitude, and longitude were registered with a high precision sub-meters GPS MobileMapper v.10 (Westminder, CO, USA). Relationships between *P. caesium* size categories (seedling, juvenile, non-reproductive adults, reproductive adults (1), and reproductive adults (2) per site, and environmental variable data (pH, organic matter, electrical conductivity, altitude, orientation, latitude, and longitude) were analyzed by canonical correlation analysis (CCorrA) and Wilk's lambda tests to identify the significant parameters in the canonical relationships. Furthermore, a Pearson correlation analysis was performed to test which CCorrA correlations were significant ($p < 0.05$). Within the significant results, the correlation coefficient r was interpreted as a strong correlation r ($\geq |0.7|$) or moderate correlation ($|0.5| \leq r \leq |0.7|$) [42]. In addition, PAST software [43] was used to assess statistically significant differences in the abundance of individuals concerning geographic orientation.

Finally, to assess the effect of guava (*Psidium guajava* (L.) Burm.) cultivation on the soil fertility and the possible consequences for the species, a soil analysis was carried out with the following characteristics: (a) cliffs with the presence of *P. caesium* without disturbance, (b) cliffs with the presence of *P. caesium* and guava cultivation on the upper slopes, and (c) cliffs with a historical presence of *P. caesium* and abandoned cultivation ($\approx$20 years) on the upper slopes. Three sites with similar characteristics (a, b, and c) were analyzed, and from each site, three soil samples ($\approx$1 kg per sample) were obtained to analyze soil fertility (nitrogen (N), phosphorus (P), potassium (K), iron (Fe), and manganese (Mn) content) using DTPA and atomic absorption. The pH, EM, and OM were obtained as described above.

A one-way analysis of variance with Tukey contrast tests ($\alpha = 0.05$) was performed to determine the significant differences in the physicochemical properties between sites using PAST [43].

## 3. Results
### 3.1. LULC Sampling Procedures and Deforestation Rate

The study had a high overall accuracy and kappa indices on the Landsat-based LULC maps for 2003 (89%, 0.84) and 2013 (91%, 0.87), respectively (Table S2). The vegetation

changes in the Non Forest category were mainly used for agriculture and pasture 160 (97%) and (3%) dams land. In the analysis, the dry tropical area studied showed a severe disturbance, with only residual forest fragments because the surface had undergone a historical change in land use towards agricultural activities (guava cultivation), and livestock. Even so, between 2003–2013, the residual subtropical vegetation decreased from 1085.04 ha to 948.21 ha (almost 13%). Furthermore, comparisons among the three heterogeneity metrics (Shannon diversity, Simpson diversity, and mean forest size) revealed no significant differences as a function of the landscape extent (500 m, 1 km, or 1.5 km), with correlations among metrics ranging from 0.85 to 0.94. However, the mean forest slightly increased when the buffer was higher than its size (Table S3).

A buffer of 1 km was selected for the study due to the Simpson and Shannon diversity indexes, which showed higher values in four of the six sites (Table S3). Results suggested that Presa Cebolletas, Barranca Tortugas, and Barranca Montoro had the main woody vegetation losses (45.65, 37.52, and 24.52 ha, respectively), while Presa Malpaso, Río Gil, and Puente Cuates lost less than 15 ha. A similar trend was recorded among sites for the annual deforestation rate (Table S4). On average, in 2003 and 2013, 22.8 ± 16.1 ha were lost in all of the sites; also, the sites showed a mean annual deforestation rate of 0.01 (Table S4; Figure 2).

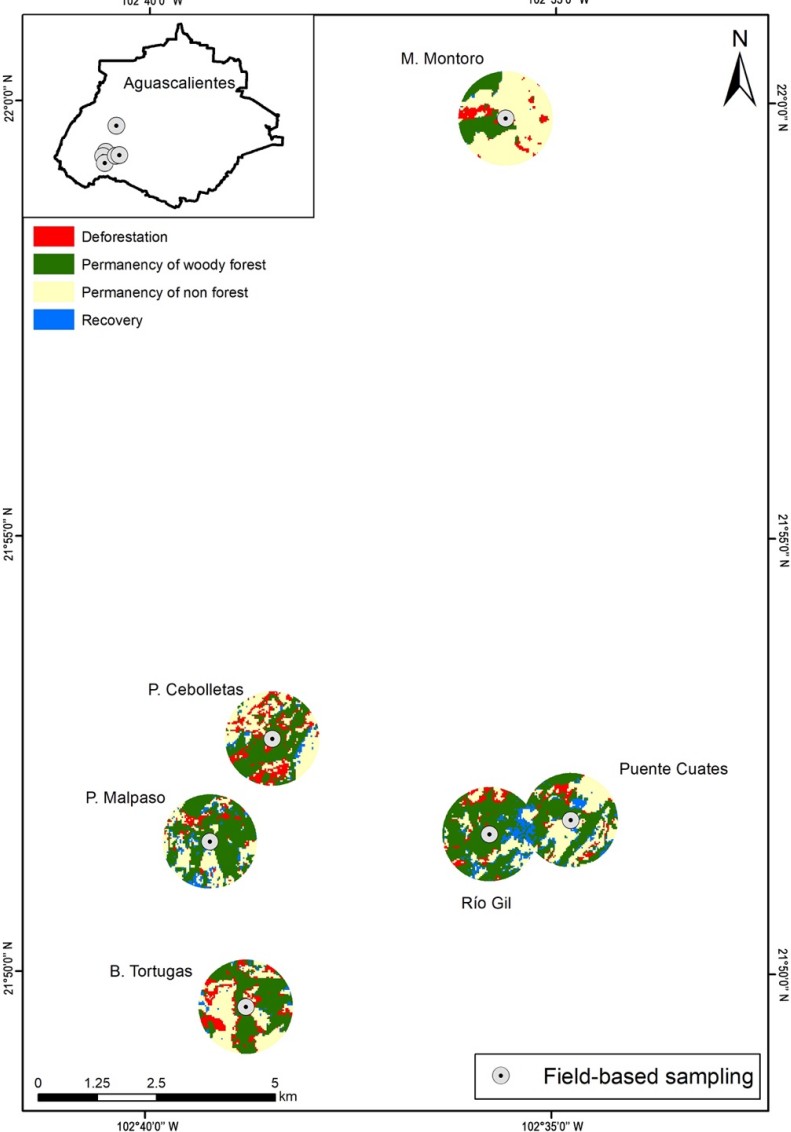

**Figure 2.** Transition changes of forest cover in buffer zones of 1 km for the period 2003–2013 in the sampling sites in the Aguascalientes State.

### 3.2. Fragmentation Metrics

Comparing the six sampled sites result between 2003–2013, only the Presa Cebolletas landscape showed an evident fragmentation process; due to this, there was an increase in PD/ED and a decrease of CPLAND (Table S5). In the case of Barranca Tortuga, Barranca Montoro, and Puente Cuates, where there was an increase of PD, however in some cases, the ED and CPLAN values were maintained or increased for this reason. In Presa Malpaso and Río Gil, the results showed that between 2003 and 2013, the landscape was preserved (Table S5).

### 3.3. Ecological Distribution, Abundance, and Size Structure

From the 220 sampled sites, *Pachyphytum caesium* only occurred at six sites (2.7%), all of them were in the state of Aguascalientes, Mexico (Figure 1). *P. caesium* was not found in four sites (Presa los Serna, Barranca Tortugas, and two sites on Río Gil) where in the past its occurrence was reported in 1995. Thus, the species ecological distribution is narrowed to remnants of the dry tropical and temperate forest of the municipality of Calvillo at an average altitude of 1932 masl. The results showed that there were significant differences in the abundance and density of individuals among plots of each site ($\chi^2$ = 21.42, $p < 0.05$) and between sites ($\chi^2$ = 22.35, $p < 0.05$) (Figure 3). At all sites, the number of individuals showed significant differences between size categories (Figure 3).

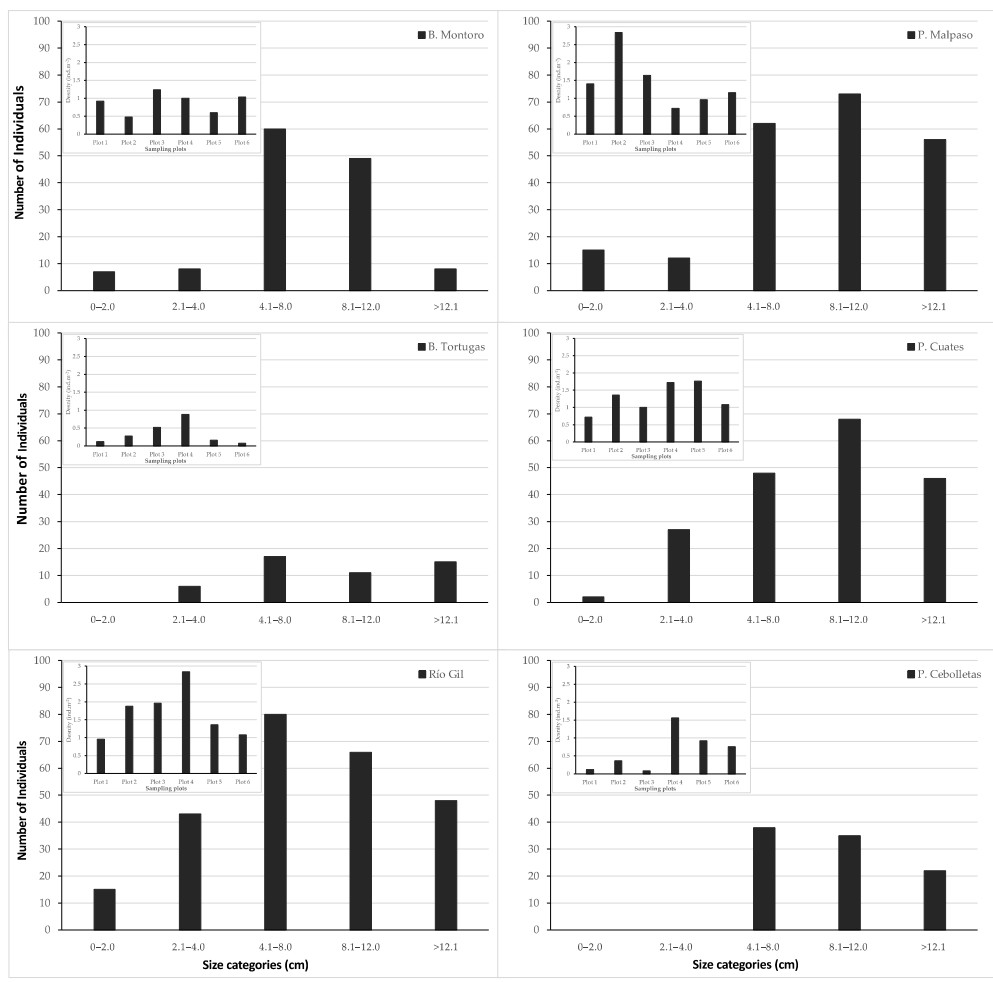

**Figure 3.** Population size structure, and density (embedded graph) per sampling site.

The population of Barranca Tortuga (0.34 ± 0.28 ind m$^{-2}$), Barranca Montoro (0.88 ± 0.26 ind m$^{-2}$), and Presa Cebolletas (0.63 ± 0.51 ind m$^{-2}$) showed a lower density compared with the sites of Presa Malpaso (1.45 ± 0.68 ind m$^{-2}$), Puente Cuates

(1.27 ± 0.38 ind m$^{-2}$) and Río Gil (1.68 ± 0.64 ind m$^{-2}$). These three populations showed more remarkable cover change and an annual rate of deforestation than Presa Malpaso, Puente Cuates, and Río Gil. In addition, population abundance was correlated with the rates of deforestation, showing a moderate correlation of 0.86.

On the other hand, three sites presented evidence of rock climbing (e.g., lost arrow piton, copperheads, rivets, fixed parabolts, or rock damage). These are Barranca Montoro, Presa Cebolletas, and Barranca Tortugas, where sport climbing negatively affected the size of the individuals due to the stem fragmentation or by killing them during the climbing process. Further, the density (ind m$^{-2}$) was significantly lower ($\chi^2$ = 25.23, $p < 0.05$) between sites where rock climbing is practiced (0.14 ± 0.11, 0.07 ± 0.04 and 0.10 ± 0.08, respectively) than in sites without evidence of climbing (0.8 ± 0.27, 0.69 ± 0.231 and 0.65 ± 0.21, respectively).

On the heatmap, weak correlations between the variables are blue, while the strongest correlations are shown in red (Figure 4). The dendrogram differentiated two clades, the first integrated by reproductive adults (1 and 2), while the second separated two more clades. The first showed an association between seedling and non-reproductive adults correlated with the juvenile categories. The boxplot (on the right) showed a negative average in seedlings, juveniles, and reproductive adults (1 and 2) due to the absence or low abundance of individuals in some populations. The non-reproductive adult category showed a positive average and a great dispersion of data due to populations with few or large numbers of individuals, in contrast with seedlings, which did not show variation due to the low density of individuals in the populations (Figure 4).

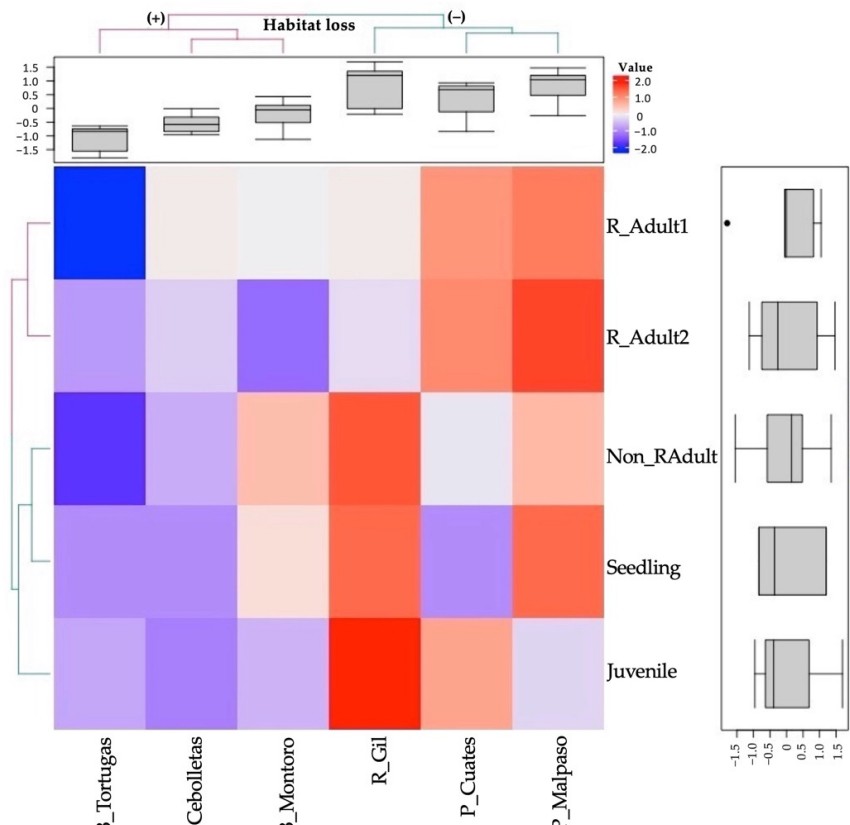

**Figure 4.** Heatmap based on the normalized abundance value of the size categories (in rows, Seedling, Juvenile, Non_RAdult: non-reproductive adult, R_Adult1: reproductive adults (1) and R_Adult2: reproductive adults (2)) of the analyzed sites (in columns, B_Tortugas: Barranca Tortugas, Cebolletas: Presa Cebolletas, B_Montoro: Barranca Montoro, R_Gil: Río Gil, P_Cuates: Puente Cuates and P_Malpaso: Presa Malpaso). The red color means the maximum correlation value and decreases towards the blue color. Populations with high (+) and low (−) levels of habitat loss.

The dendrogram from the heatmap is clearly divided the samples into two groups (Figure 3). The first group consisted of the sites with the most significant disturbance caused by rappelling and forest clearing (Barranca Tortugas, Presa Cebolletas, and Barranca Montoro) with lower abundance values (see the upper boxplot). The second cluster grouped three sites with lower disturbance (Presa Malpaso, Río Gil, and Puente Cuates), with the most abundant populations (see the upper boxplot).

*3.4. Environmental Characterization of Sites and Soil Analysis*

Although the species showed a restricted distribution to rocky cliffs, it was also found in steep hillsides (shelf with slope angles <90° with heights between 4 to 10 m). A significant difference in orientation was observed among sites (Table S6). Barranca Montoro, Puente Cuates, and Tortugas averaged southeast-facing slopes, while Río Gil was oriented to the southwest. Hillsides of Presa Cebolletas and Presa Malpaso showed a northeast and northwest orientation, respectively.

CCorA results showed that *Pachyphytum caesium* size category abundance is affected by environmental variables (Figure S1). The Wilks lambda on F1 was 0.032, indicating a high degree of certainty for the two data sets. The abundance and size (1 and 2) of reproductive adults in the first and non-reproductive adults in the second quadrant were strongly affected by OM, slope orientation, and latitude; these categories were also affected by pH and longitude in the fourth quadrant. Additionally, it was also observed that EM and altitude positively influenced these categories. However, the opposite happened in seedlings and juvenile categories, favored by the OM, orientation and pH and affected by the EM and altitude (Figure S1).

Finally, the results for soil properties in sites with and without *Pachyphytum caesium* are shown in Table S7. Significant differences in soil ($p < 0.05$) were found between areas with guava cultivation on the upper slopes and undisturbed sites. Also, soil properties (except pH and nitrogen) changed significantly on hillsides with guava cultivation or abandoned areas, and these soil properties showed significant effects in *P. caesium* abundance and presence. All of these results suggested that the species can be classified as PSESP.

**4. Discussion**

The distribution of *Pachyphytum caesium* is restricted to cliffs and canyons habitats of the dry tropical forest and transition toward the temperate forest of Aguascalientes. In this study, we expected to find it in neighboring states (Jalisco and Zacatecas) as they present these habitats and types of vegetation [44]. However, our study showed that this species is endemic, with a fragmented distribution. This result is similar to those found in other species of the genus *Pachyphytum* [20,45] and the Crassulaceae family [19]. These similarities among species distributions can result from natural events [4] or recent anthropogenic disturbances [23] exacerbated by the reduction and fragmentation of tropical dry forests in central Mexico, primarily Aguascalientes.

Accelerated climate change and land-use change are two of the major threats affecting species of the Crassulaceae, because most of their populations are small and considered threatened [19,45,46]. *Pachyphytum caesium* is highly sensitive to extreme environmental changes like other rare species (e.g., [4,47]). It is known that two *P. caesium* populations (Presa Los Serna and Barranca Tortuga), collected in 1994 by Eduardo García (housed at the Autonomous University of Aguascalientes Herbarium), disappeared following heavy hail and frost that occurred in the 1990s, a fact corroborated by this study. *P. caesium* is probably very vulnerable to climate change and local changes in environmental conditions as other species (e.g., [4,47,48]), changes that have unfortunately become common in recent decades. For this reason, in the future, it will be necessary to model the effect of climate change on the species, with an emphasis on the microclimatic conditions of each cliff, which is often disregarded in macroclimatic predictive models.

On the other hand, the process of landscape change as a result of habitat loss and fragmentation processes recorded in the study area are consistent with Siqueiros-Delgado et al. [23],

who suggest that the conversion of land in the dry tropical forest of Aguascalientes, for agriculture and livestock, exceeds 90% of the original area. This landscape change not only affects the population abundance and occurrence of *P. caesium*, but it can also affect their population structure, as suggested by Clark-Tapia and Quintero [46]. This change is probably involved in the local extinction of two populations reported in 1995 and not found in this study. It has been documented that agriculture often leads to a deterioration of the land in dry tropical forests that causes, among other things, erosion or modification of the biogeochemical cycles [49–51]. Although these changes are reported in different tropical landscapes, the result of soil parameters suggests a negative change that affects the habitat of *Pachyphytum caesium*. This result could be due to fertilizer runoff that is applied in guava agriculture practiced on cliff platforms. This soil chemistry modification causes a decrease of individuals and could lead to the local extinction of the population; however, more soil analyses will be necessary to reaffirm this hypothesis. Despite this, several studies have shown that any change in the nutrients and pH of the cliff soil is crucial, as it directly affects plant populations [14,52,53], as with *P. caesium*.

Since the 2000s, rock climbing is increasing in popularity in the cliffs and canyon areas where *Pachyphytum caesium* has been distributed. This activity drastically affects cliff-dwellers population in terms of their structure and composition [16,54–56]. Rock climbers cause soil loss, which prevents the establishment of new plant individuals, modifying their population dynamics [16,55,56]. Our results showed that sport climbing adversely affects the populations from Barranca Montoro, Presa Cebolletas, and Barranca Tortugas. Our field observations, followed by the analysis implemented, showed that rock climbing reduces the number of individuals and affects the plant sizes and the amount of soil or substrate. This result suggests that rock climbers need to be aware of the natural inhabitants to diminish the impact on cliff-dwelling such as *P. caesium*. Otherwise, damage to the cliff populations can be severe enough to cause its local extinction in a short period of time. Although this study was able to estimate the effect of rock climbing, it is still necessary to implement regulations; for example, to establish climbing routes, specify the frequency each path is in use; establish the number of climbers per route; and designate the different parts of the cliffs that climbers should avoid. These regulations will help to ameliorate the risk of plant mortality principally in small individuals, as well as the considerable loss of soil substrate.

On the other hand, abundance differences between plots and sites suggest that the microhabitat resources act on the growth and size of the plants [54]; also, the disturbance of this area implies the loss of this specific habitat [53,54]. For this reason, it is essential to understand which abiotic and anthropogenic factors are associated with the loss of abundance and distribution of *Pachyphytum caesium*, but also which size categories of the plants are more vulnerable to habitat disturbance [57], especially on PSESP (e.g., [58–60]). In *P. caesium*, any change in its specific ecological niche (e.g., [15,61,62]) would significantly affect their abundance and distribution. For example, changes in soil availability and nutrient for seedlings roots and grow [52], as well as pH requirements [15,52], temperature, light, or humidity [61,62], in addition to other factors such as altitude, latitude, and orientation [15,63]. This cliff species is highly vulnerable to local extinction as habitat changes or anthropogenic disturbances negatively affect population growth. Recent demographic records suggest that *P. caesium* populations in Barranca Tortugas, Presa Cebolletas, and Barranca Montoro have been reduced by up to 50% due to the introduction of guava cultivation, rock-climbing, or environmental factors (frosts and hailstorms) (Clark-Tapia unpublished results). At an ecological level, this could cause low sexual recruitment in *P. caesium* populations. This recruitment loss could potentially lead to genetic structuration at a genetic level, thereby increasing the number of risk factors and the vulnerability of plant species with extremely small populations [7,8].

This research is essential as it represents the first ecological study of one of the species of the *Pachyphytum* family and could be a reference to understand the ecology of phylogenetically close species under conditions similar to those reported in this study. For

example, our results showed that *Pachyphytum caesium* can be classified as PSESP due to their specialized habitat, restricted distribution, and high vulnerability to habitat loss and environmental change, which threatens their populations and, makes it more prone to extinction [6,7,9]. Therefore, it is recommended as an immediate strategy to incorporate the species into the Mexico federal protection list (NOM-059-SEMARNAT-2010) as a threatened species according to its methodology (Risk Assessment Method). In addition to the regulatory proposal, a tissue culture project of *P. caesium* must be carried out to rescue and plant conservation, and a public awareness program should be implemented among climbers, as these are viable conservation strategies for threatened species, as suggested by Croteau et al. [64]. Finally, in other Crassulaceae species, a high limitation of pollinators, a limited pollen movement, and many vain seeds have been reported [28,29]. All these limitations are likely to be present in *P. caesium*; thus, we recommend realizing future genetic and reproductive studies with this species in order to understand its biology and provide the best conservation recommendation.

## 5. Conclusions

Our results showed that *Pachyphytum caesium* is an endemic species of Aguascalientes, and it can be classified as a PSESP because its specialized habitat, restricted distribution, small population, and high vulnerability to habitat loss and environmental change, which threatens their populations. Considering the above as well as the diverse social-economic impacts exerted on its populations, we consider that *P. caesium* deserves a status of being threatened in the Mexico federal protection list (NOM-059-SEMARNAT-2010).

**Supplementary Materials:** The following are available online at https://www.mdpi.com/article/10.3390/d13090421/s1. Figure S1: Canonical correlation analysis (CCorA) biplot showing the relationship between the environmental variables (electrical conductivity (EM), organic matter (OM), pH, orientation, altitude, latitude, and longitude) and size categories at the sampling sites. Table S1: Description of sampling sites in Aguascalientes. The average characteristics of altitude and orientation are described, as well as the climate and dominant vegetation type. The five dominant species at each site are ordered based on their abundance. Table S2: (a) Confusion matrix between classes for the 2003 image and (b) confusion matrix between classes for the 2013 image. Table S3: Comparisons of mean size of forest, Shannon diversity index, and Simpson diversity index to evaluate which buffer size (500 m, 1 km, or 1.5 km) helps to better describe the landscape. In bold are the highest values. Table S4: Annual deforestation rates (%D) and area covered by the forest (ha) from 2003 to 2013 within the buffer of 1 km of each site. Area of forest cover for the initial (A1) and final (A2) year during each transition period is shown. Table S5: Changes in fragmentation (patch density (PD), edge density (ED) and core percentage of landscape (CPLAND) for 2003–2013. Table S6: Orientation of sampling sites. Circular mean which takes circularity of the data, and 95% confidence interval of the mean using 5000 bootstrap replicates (IC). R: Rayleigh's test for uniform distribution and significance *p*-value that was obtained. Table S7: Mean values of the soil parameters evaluated on the cliffs of Aguascalientes. All of the variables were measured for cliffs with the presence of *P. caesium* without disturbance (WD); with the presence of *P. caesium* and guava cultivation on the upper slopes (w Pc-G); and cliffs with the historical occurrence of the species and abandoned areas (without Pc-G). Organic matter percentage (OM), potential of hydrogen (pH), nitrogen (N), phosphorus (P), potassium (K), iron (Fe), and manganese (Mn). The standard deviation is given in parentheses.

**Author Contributions:** R.C.-T., G.G.-A. and C.A.-C. conceived the idea, and performed the research and collected the data; R.C.-T., V.A.-H., N.P.-C. and J.J.V.T.U. analyzed the data; R.C.-T., C.A.-C., V.A.-H., J.E.C., S.C.-L. and L.I.P.-G. wrote the first draft of the manuscript and all of the authors contributed substantially to the revisions. All authors have read and agreed to the published version of the manuscript.

**Funding:** This research received no external funding.

**Institutional Review Board Statement:** Not applicable.

**Data Availability Statement:** The data presented in this study are available in the main text and within the supplementary material section.

**Acknowledgments:** The authors acknowledge the support of the Universidad Autónoma de Aguascalientes and the Universidad de la Sierra Juárez (CAUP-2-EA-0714).

**Conflicts of Interest:** The authors declare no conflict of interest.

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
