# Peer review of "Effects of Habitat Loss on the Ecology of Pachyphytum caesium (Crassulaceae), a Specialized Cliff-Dwelling Endemic Species in Central Mexico"

_diversity, doi:10.3390/d13090421_

Round 1

Reviewer 1 Report

Clark-Tapia et al. present interesting results about the state of a threatened cliff-dwelling species. This is a timely and important report, with clear theoretical and applied implications. Apart from a few minor issues, the science is sound. However, the quality of writing is a little problematic, and some “polishing” of paragraph and sentence structure is essential.

Major comments:

L 50: Are all endemic or globally threatened species also PSESPs and vice versa?

L 100-104: These are wide ranges. Are there any geographic trends (e.g., aridity increases northwards)?

L 177-178: Do plants <8 cm do not reproduce and plants >8 cm reproduce? Is this an actual phenological threshold, or just a size category that is roughly associated with life stage?

L 384-395: You could have taken soil samples in two sites where the species has become extinct, to demonstrate agriculture’s negative effect. I suspect you could find that soil properties are more deteriorated than shown in sites where the population is in decline, thus validating what you suggest here.

L 432-435: You already defined it as an SPESP in the Introduction. This can’t be both background knowledge and a conclusion.

Minor comments:

L 22-23: I recommend replacing this “banal” and general sentence with one about the importance of cliff-dwelling or small population plants.

L 29-33: A few minor repetitions (e.g., effect of rock climbing)

L 35: Rock climbing is not habitat loss, maybe habitat degradation, or better yet “disturbance”.

L 43-47: The structure doesn’t flow to well.

L 47-49: Unclear, both as a statement of its own, and in reference to previous sentences.

L 60-74: Not well organized

L 83: “at human convenience”?

L 129-140: What is the resolution?

Tables 1-3: I would move these to the Supporting Information, but bring Fig. S2 into the main text.

L 300-304: I do not understand how this fits with the size categories being pre-determined (L 177-178).

L 322-341: This part feels a little too long, and not focused on the main findings.

L 323-326: Correlations were not significant, apart from those that were.

L 382-384: What?

L 400-403: Refer to tables/figures.

L 424: “negatively”? I am not sure about the type of effect when referring to variables like orientation.

Author Response

We thank the reviwer's carefully reading and signification comments. Considering your comments we made the following responses (see file attached). Revised portions are marked in yellow in the paper.

Reviewer 2 Report

The authors have presented an interesting work, which, in my opinion, is worth of publication. Some miror comments/suggestions are found below. Hopefully, they are useful to the authors.

1. All plant scientific names should be written in italics. Please correct accordingly (e.g., lines 120, 200, 213, 214, 215, 340, 587, etc).

2. How did you select the 220 sampling sites of Pachypodium? Were these sites selected to include at least some habitats favoured by Pachypodium (i.e., they were selected to represent, to the best of your knowledge, the preferred habitats of Pachypodium, rocky cliffs, steep hillsides)? If not, then it is expected that several sampling sites will not contain any Pachypodium As a consequence, the plant may appear rarer than it actually is.

3. Please provide us with some details on how you performed the experimental field work on the Pachypodium Given that Pachypodium grows on steep hillsides and cliffs, did you use climbing techniques to reach the populations and make counts of the plants and assignment into size categories?

4. Lines 358-359: are there recent or historical records of Pachypodium from other states apart Aguascalientes? If yes, it would be better not to consider it an Aguascalientes endemic.

5. Lines 417-419: I have some difficulties to understand the meaning. Perhaps rephrase the sentence?

6. Line 439: Do you think that tissue culture (i.e., production of plants originating from one or very few parental individuals) would be a good idea to produce offspring and re-introduce plants into the wild? This could be a bad decision, particularly in the case that outcrossing is necessary to produce viable seed (compare lines 126-127, 441-443) and further, could have a negative impact on genetic diversity.

Author Response

We thank the reviewer carefully reading and signification comments. Considering your comments we made the following responses (see file attached). The revised change are made in yellow in the article.

Reviewer 3 Report

Dear Authors,

The submited manuscript titled „Effects of habitat loss on the ecology of Pachyphytum caesium  (Crassulaceae), a specialized cliff-dwelling endemic species in  central Mexico” provides very interesting results and might interest large international audience. The manuscript is generally well-written but I have found some flawns, which- in my opinion- should be improved.

  1. The title is too general due to use the term „ecology”. In context of presented findings I think that more suitable title should by e.g. „ The effect of habitat loss on abundance and structure of populations Pachyphytum caesium  (Crassulaceae)- a specialized cliff-dwelling endemic species in  central Mexico”.
  2. Line 169-In my opinion there is lack of explanation of choice sites, where detailed investigations of population features were performed (Tortugas; Montoro; Río  Gil; Malpaso; Cebolletas and Puerto Cuates). What were the differences amnong localities? Additionally the table containing comparation of habitat conditions (e.g. elevation a.s.l., plant cover, list of dominant species in community or acompaining species, light intensity at ground level) would be very useful.
  3. Line 170- how the study plots were chosen (randomly or systematically), perhaps the figure with sketch of plot distribution might be added.
  4. In chapter „Material and methods” I suggest to add one subchapter titled „The statistical analyses” containing description of use all statistical tests.
  5. Lines 260-276-I encorage Authors to present data reffering to abundance and density of populations on chart or in table. Moreover, the age structure of populations (containing share of seedlings, juvenils, vegetative and generative adults) would be very useful in comparison of populations.
  6. In chapter „Discussion” the perspectives of persistence of studied populations in occupied sites should be highlighted. Perhaps some ways of protection this rare species might be proposed.
  7. In my opinion the chapter „Conclusions” should be added.

Author Response

We thank the reviewer carefully reading and significative comments. Consider your comments we made the following responses.

Round 2

Reviewer 1 Report

I have read the revised manuscript with quite delight. It is undoubtedly much improved, and nearly ready for publication. Nevertheless, I feel that a few minor language corrections are still required.

L 22: “Cliff-dwelling plant species are highly specialized and adapted to […], and are mostly endemic, narrowly-distributed and threatened.”

L 34-37: I think these should be merged into a single sentence (or two).

L 44: Change “worst” to “critical”, “acute”, “alarming”, etc.

L 46:”characterized by” is redundant.

L 48: Change “also” to “hence” or “consequently”.

L 53-54: “Survival of PSESP inhabiting … is more sensitive to the presence …”

L 102-105: This is better, however: (1) the second sentence is not really a sentence, (2) it will far better to see these climates on the map itself.

Reviewer 3 Report

Dear Authors,

Your manuscript received sufficient improvement, therefore I do not have any further suggestions and remarks.